# Consistent tumorigenesis with self-assembled hydrogels enables high-powered murine cancer studies

Abigail K. Grosskopf [1], Santiago Correa[2], Julie Baillet [2], Caitlin L. Maikawa [3], Emily C. Gale [4], Ryanne A. Brown [5] & Eric A. Appel [2,3,6,7 ✉]

Preclinical cancer research is heavily dependent on allograft and xenograft models, but current approaches to tumor inoculation yield inconsistent tumor formation and growth, ultimately wasting valuable resources (e.g., animals, time, and money) and limiting experimental progress. Here we demonstrate a method for tumor inoculation using self-assembled hydrogels to reliably generate tumors with low variance in growth. The observed reduction in model variance enables smaller animal cohorts, improved effect observation and higher powered studies.

[1] Department of Chemical Engineering, Stanford University, Stanford, CA, USA. [2] Department of Materials Science and Engineering, Stanford University, Stanford, CA, USA. [3] Department of Bioengineering, Stanford University, Stanford, CA, USA. [4] Department of Biochemistry, Stanford University, Stanford, CA, USA. [5] Department of Pathology, Stanford University School of Medicine, Stanford, CA, USA. [6] Department of Pediatrics—Endocrinology, Stanford University, Stanford, CA, USA. [7] ChEM-H Institute, Stanford University, Stanford, CA, USA. ✉email: eappel@stanford.edu

A reproducibility crisis in preclinical research has contributed to disappointing outcomes in clinical trials[1–3]. In response, significant effort has gone into developing animal models that better recapitulate human disease with the hope of improving the predictive power of preclinical research[4,5]. Yet, little attention has been paid to improving consistency of animal models or developing facile, lab transferable techniques[6]. As a result, inconsistent animal models are endemic and complicate comparisons of data from different labs, within labs, and even within individual experiments[7]. Subcutaneous allograft and xenograft flank models are among the most common models used for preclinical cancer research, particularly for immunooncology[8]. These models involve injecting cancerous cells dispersed in liquid buffered saline subcutaneously on the flank of animals and waiting for tumors to form[9] (Fig. 1a).

Variance in tumor formation necessitates researchers use more animals to conduct sufficiently powered studies, and the chance of failed tumor formation further inflates animal cohort sizes (Fig. 1a). Indeed, technical challenges typically lead to upwards of 30% of inoculated mice failing to form tumors[10]. Preclinical cancer studies can involve hundreds of mice, so significant resources, time, and animals can be wasted due to the severe inefficiencies present in current model protocols. Since mice with abnormal or late-growing tumors must be euthanized without having contributed useful data to a study, significant ethical concerns are raised by the unnecessary overuse of research animals. Reliable and reproducible models would dramatically reduce the number of mice needed and increase the rate at which discoveries can be made by improving study power.

Current approaches to cancer cell inoculation suffer from two primary drawbacks: (i) inoculation is poorly reproducible animal-to-animal because of cell agglomeration or settling in the syringe prior to injection, and (ii) implanted cells lack extracellular matrix providing critical biochemical and biophysical cues[11]. Previous reports indicate that cancer cells encapsulated in basement membrane extract (BME, tradenamed Matrigel®) form tumors more rapidly in murine models when compared to cells suspended in saline[10,12,13]. While these results are promising, BME is a poorly defined solution derived from Engelbreth-Holm-Swarm mouse sarcoma, thereby suffering from considerable batch-to-batch variability, and its temperature-dependent gelation introduces technical difficulties in avoiding premature gelation within the syringe[14]. While BME offers tumorigenic factors like Laminin and Type IV Collagen, the uncertainty of its contents and technically challenging handling provide good reason for researchers to avoid it. There has been increasing interest in the development of alternative, molecularly defined hydrogel scaffolds for applications in tissue engineering[15]. Injectable hydrogels have been developed that can enhance therapeutic cell administration by protecting cells from mechanical forces during injection, enabling homogeneous injections and enhancing cell retention at the injection site[16,17]. Here we develop a self-assembled hydrogel for controlled encapsulation and delivery of cancer cells that improves the reproducibility of tumor formation.

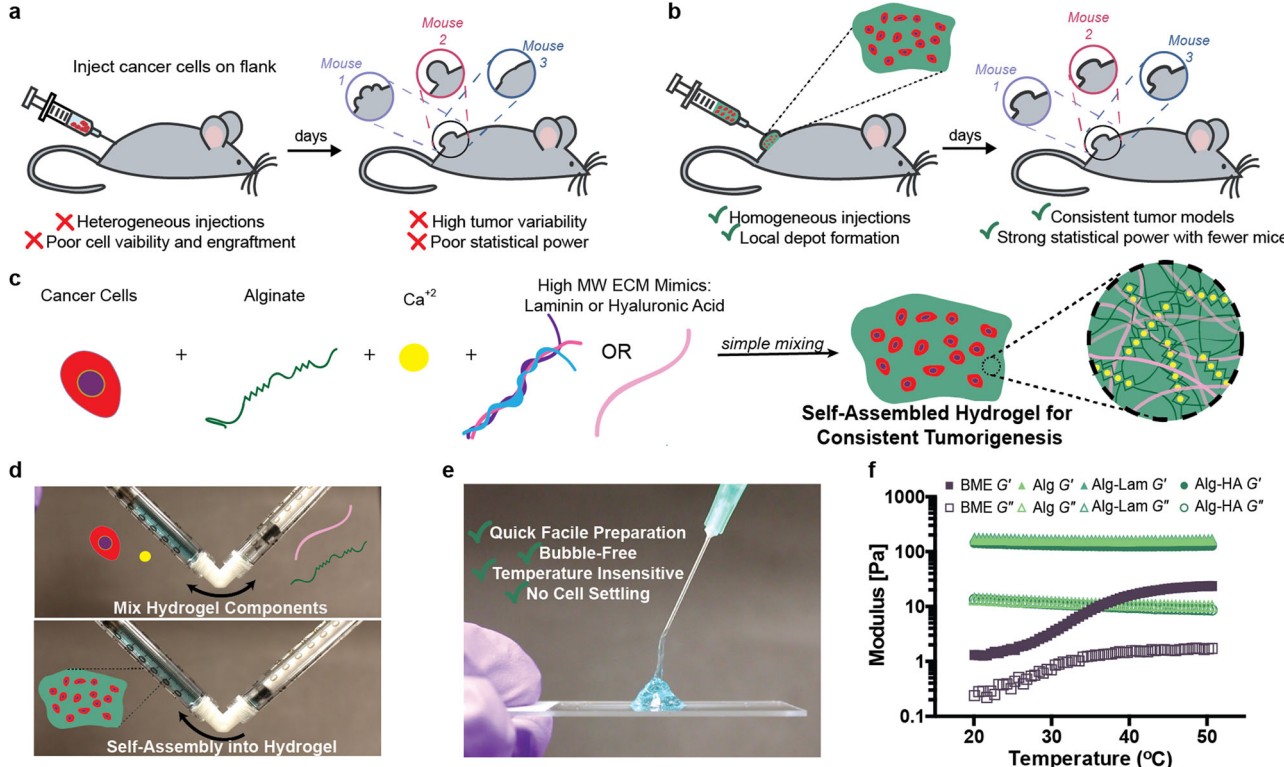

**Fig. 1 Self-assembled hydrogels for preclinical tumor models. a** Current method of forming syngeneic subcutaneous tumors by injecting cancerous cells in liquid saline. **b** Encapsulation process of B16F10 cells in self-assembled hydrogels and the benefits of cell delivery using hydrogels. **c** Components that form self-assembled hydrogels including cancer cells, alginate, calcium, and extracellular matrix-mimicking biopolymers. **d** Dual syringe mixing strategy to formulate self-assembled hydrogels and encapsulate cells. A polymer solution containing alginate and ECM additives is loaded in one syringe (right) and cells and calcium sulfate are loaded into the second syringe (left). Blue dye is included to aid visualization for this demonstration. The two solutions are mixed with a female–female dual-syringe mixer by pumping the material back and forth for 30 s to form a robust hydrogel pre-loaded into a syringe ready for administration. **e** Injection of self-assembled hydrogels into a hydrogel depot through syringe needles (25G). **f** Oscillatory Shear Temperature ramp ($w = 10$ rad/s, $y = 1\%$).

We validate our methods with the widely used B16F10 melanoma model (Fig. 1b).

## Results

We use alginate-based hydrogels due to great biocompatibility, and their mild and rapid formation by simple mixing with calcium (Fig. 1c)[18,19]. This facile fabrication method does not involve complex chemical reactions affording efficient encapsulation of viable cancer cells. These cell-encapsulated injectable hydrogels quickly self-assemble with 30 s of mild mixing of two precursor solutions (e.g., alginate and calcium) in a dual-syringe setup that results in the hydrogels pre-loaded in a syringe and ready for injection (Fig. 1d, e and Supplementary Videos 1 and 2). Further, these alginate-based hydrogel formulations can be easily supplemented with laminin (Alg–Lam) and hyaluronic acid (Alg–HA), which are components of the ECM in endogenous tumors[20,21]. All alginate-based hydrogel formulations display temperature-insensitive dynamic mechanical properties (Fig. 1f and Supplementary Fig. 1). In contrast, BME exhibits irreversible temperature-dependent gelation and much weaker dynamic mechanical properties. For our studies, laminin and HA were added in the maximum quantities possible to formulate consistent alginate-based hydrogels and maintain injectability. Short-term cell viability studies in two commonly used cancer cell lines indicate that the hydrogel formulations are not cytotoxic and that the addition of adhesion motifs such as laminin and HA improve cell growth (Supplementary Fig. 2). Different cell lines interestingly demonstrated variable responses to growing in BME.

To demonstrate our method, Luc$^+$B16F10 cancer cells were encapsulated in 50 μL of hydrogel formulations and injected subcutaneously on the flank of C57BL/6 mice through a 21G needle[9]. Tumor growth was compared with tumors administered in 50 μL of BME and saline. Tumor formation was observed for the first 10 days with in vivo imaging (Supplementary Figs. 3 and 4). Tumors established with saline displayed lower signal on Day 1 than all other formulations, indicating poor cell viability and retention after inoculation, corroborating observations of inconsistent tumor formation reported previously[9]. These initial tumor growth data were additionally fit to a Gompertz model (Supplementary Fig. 5). The fits suggest that our hydrogel approach does not hinder proliferation and may improve end carrying capacity of the tumor compared to saline controls[22]. Degradation of alginate was also monitored using in vivo imaging and fluorescentlylabeled alginate. Our results demonstrated that alginate degraded over the course of 10–12 days with a half-life of approximately 4 days (Supplementary Figs. 6 and 7). This timescale of degradation aligns with the B16F10 cells growing into a robust tumor, remodeling the matrix, and degrading the alginate, suggesting there should not be significant effects of alginate being present during treatment.

Once tumors reached 100 mm$^2$, tumor area was measured over time with digital calipers (Fig. 2a–e, Supplementary Fig. 8 and Supplementary Table 1). The coefficient of variance (%CV) was quantified (Fig. 2a–e) and revealed that tumors inoculated in a saline vehicle showed the greatest variance among all of the groups evaluated. Tumors formed in BME showed the fastest growth, but also demonstrated a high %CV (40%). Alginate hydrogel alone (Alg) yielded moderate tumor growth with lower %CV (<25%), while the hydrogels comprising tumorigenic ECM components Laminin (Alg–Lam) and HA (Alg–HA) exhibited enhanced tumor growth while maintaining low %CV values (<25% and 15%, respectively) that were less than half that of tumors from saline and BME. The Alg group likely showed the slowest group due to the absence of an additive stimulating tumorigenic biomolecule, like laminin or HA.

Tumor histomorphology on day 15 post-inoculation evaluated by a blinded pathologist was comparable for tumors formed in hydrogel groups (regardless of composition), BME, and saline (Fig. 2f–j), suggesting that our delivery method does not change long-term tumor morphology and applicability of tumors formed in hydrogels for preclinical studies. Replicate representative images suggest our delivery method does not change the resulting tumor histomorphology (Supplementary Figs. 9–13). Percent necrosis analysis in each tumor and staining with CD31 by a blinded pathologist also suggested comparable angiogenesis and blood supply in all groups (Supplementary Figs. 14 and 15). Finally, an analysis of lymphocyte infiltration revealed similar numbers of lymphocytes infiltrating the tumors in all groups, except the Alg hydrogel group showed slightly reduced infiltration (Supplementary Fig. 16).

Researchers aim to start treatment when tumors are a given size (e.g., 100 mm$^2$)[9,10], which for practical reasons would preferably occur on the same day for all mice; however, treatment must be staggered if tumors grow at different rates. Indeed, 30% of the mice inoculated with cells in saline did not form tumors during the 15-day study (Fig. 3a), corroborating previously reports[10]. We therefore analyzed the effect of tumor variance on treatment day by identifying the proportion of mice treated in each group over time using interpolation to predict the day each mouse could be treated (Fig. 3b and Supplementary Fig. 17). The average treatment day and corresponding standard deviation for each formulation were also determined (Fig. 3c), corroborating the variance in tumor growth described above. Tumors inoculated with BME exhibited high variability for time-to-treatment (11.5 ± 1.5 days), highlighting that researchers would have to stagger experimental treatments continuously over the course of 3–4 days. In contrast, tumors inoculated in Alg–Lam and Alg–HA hydrogels exhibited lower variability (12.7 ± 0.9 and 11.9 ± 1.0 day, respectively), resulting in a more convenient treatment schedule and likely improving the precision of treatments.

Reduced model variance reduces type II error, allowing researchers to use fewer animals and observe differences between treatments with higher power. Assuming that the variance in tumor areas due to the model at the start of treatment is representative of the variance throughout treatment, it is possible to determine the impact of the improved model on design of sufficiently powered studies (Fig. 3d–f). The reduction in the tumor variance observed with Alg–HA hydrogels compared to saline and BME results in a dramatic reduction in the number of mice required to observe differences between treatments. We performed calculations with only best performing the Alg–HA hydrogel group and control groups to reduce the number of groups and complexity in this power analysis. For example, to observe a 30% difference between means of two treatment groups with 80% power, an Alg–HA-based B16F10 model would require approximately 80% fewer mice than a BME-based model (Fig. 3d and Supplementary Fig. 18). A reduction in model variance therefore reduces the number of mice required for sufficiently powered preclinical cancer studies. Alternatively, if researchers plan to use 10 mice in each experimental group (a typical group size[4]), an Alg-HA-based model would enable observation of significant differences between groups down to effect sizes of only 21%, whereas a BME-based model would only permit observations of effect sizes larger than 56% (Fig. 3e). The reduced variance of alginate-based tumor models can therefore enable the observation of more subtle effects and may lead researchers to important discoveries that would otherwise be missed. If researchers plan to use 10 mice per group, a decrease in model variance yields much higher powered studies. In a study evaluating a 30% effect size in 10 mice, a Alg-HA-based model enables a 60% increase in power over a BME-based model (Fig. 3f).

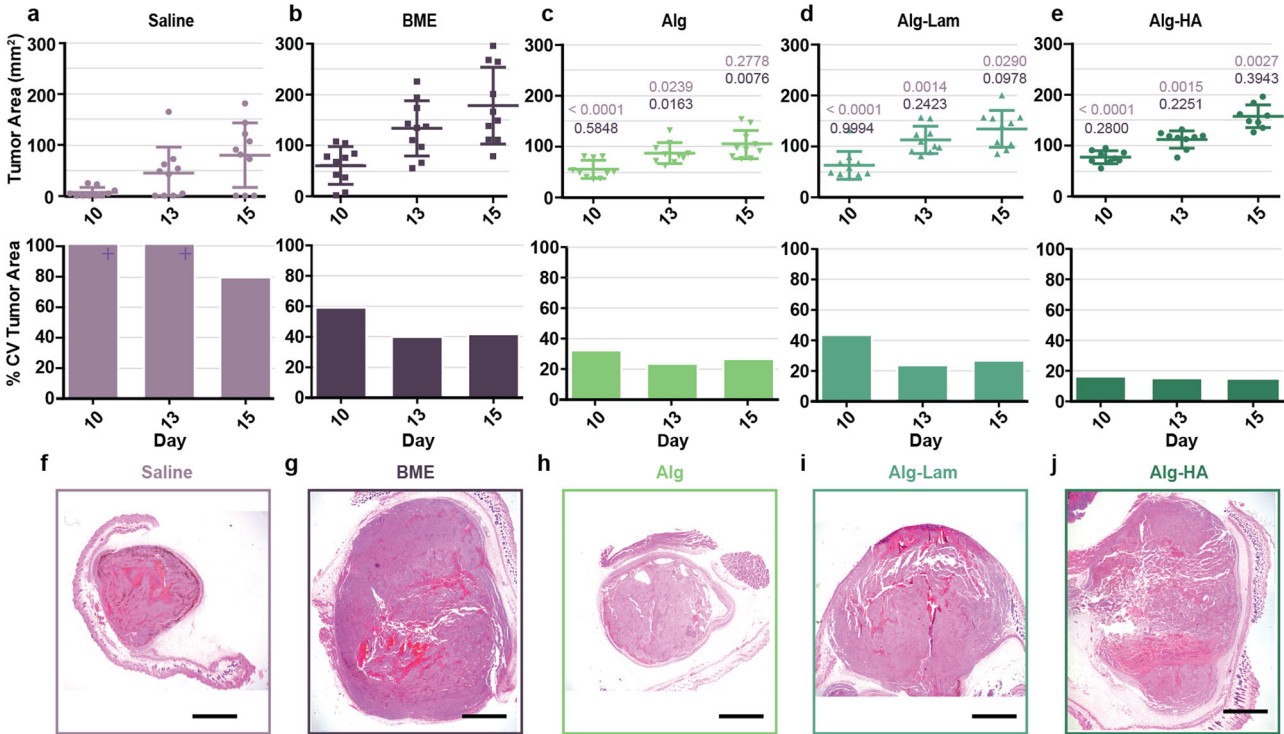

**Fig. 2 Syngeneic tumor model evaluation. a–e** Tumor area measurements over time following inoculation with various formulations containing encapsulated B16F10 cells (*n* = 9 or 10 tumors). Average tumor area and standard deviation, and the corresponding resulting %CV. Supplementary Table 1 displays all *P* values. Saline exceeded an 100 %CV on days 10 and 13. Gray lines included to guide the eye for better comparison. **f–j** Histology images of tumors excised on day 15 following inoculation with hematoxylin and eosin staining. All scale bars represent 2 mm.

Our method has the capacity to enhance the reliability of tumor formation and consistency of tumor growth for the widely used B16F10 model, demonstrating an opportunity to improve preclinical cancer models to aid observation, increase study power, and reduce resource usage (i.e., fewer mice and less researcher time). The hydrogel encapsulation procedure we demonstrate here, which only requires a simple mixing step using commercially available precursors and standard supplies, is accessible to all preclinical cancer researchers. We show improved results over current methods that utilize either saline or BME. While extensive research has focused on developing biomaterials for tissue engineering applications, the use of biomaterials to generate more reproducible in vivo cancer models has not been extensively explored. We demonstrate here, using predictive modeling, that a biomaterials-enhanced cancer model can reduce technical burden and simplify study design.

## Methods

**Hydrogel formation by self-assembly.** Hydrogels were prepared using a dual syringe mixing technique. Briefly, a stock solution of sterile Alginate (Pronova UP LVG) (5 wt%) was prepared by adding saline to the polymers and allowing them to dissolve over 1 day at 4 °C. This alginate is higher in G content, leading to reduced immunogenicity[18]. A stock solution of alginate (5 wt%) and HA (Lifecore Biomedical, 1.5 MDA, 1.25 wt%) was also prepared in saline. Biopolymer stock solutions were loaded into one syringe at the volume needed to reach the desired final concentration (1 wt% polymer total, and 10 mM calcium). A stock solution of calcium sulfate (250 mM) was prepared in water in a large container in the form of a slurry. The calcium sulfate stock solution was stirred vigorously and quickly added to an eppendorf tube containing cells and saline. Laminin Mouse Protein (Thermo Fisher) was added to the solution containing calcium and cells to a final concentration of 1 mg/mL. HA was added to a final concentration of 20 mg/mL. The stock solution of calcium and cells was then moved to a luer lock 1 mL syringe. The syringes containing the two stock solutions were then connected using a female–female luer lock mixer and the gels were prepared by mixing for 30 pumps. Once mixed, the cell-loaded hydrogels were pushed into one of the syringes, which was then removed from the leur lock mixer and equipped with a needle for application.

**Conjugation of Cyanine7 fluorescent dye to alginate.** Fluorescent alginate was prepared using carbodiimide chemistry according to established protocols[23,24] whereby Sulfo-Cyanine7 amine (5 mg, 0.0062 mmol; Lumiprobe) was dissolved under stirring in 15 mL of an alginate solution (10 mg/mL) formed in 0.1 M MES buffer at pH 6 (Thermo Fisher). Sulfo-NHS (41 mg, 0.19 mmol; Biovision) and 1-ethyl-3-(3-dimethylaminopropyl)carbodiimide (EDC; 72.5 mg, 0.38 mmol; Sigma-Aldrich) were successively added, and the reaction mixture was stirred at room temperature for 20 h. The crude product was dialyzed against deionized water for 3 days (3.5 kDa MWCO) and lyophilized until dried.

**Rheological characterization.** Rheological testing was performed using a 20 mm diameter serrated parallel plate at a 750 μm gap on a stress-controlled TA Instruments DHR-2 rheometer. All experiments were performed at 37 °C to be representative of physiological conditions. Frequency sweeps were performed at a strain of 1%. Temperature sweeps were performed at a strain of 1% and a frequency of 10 rad/s. Amplitude sweeps were performed at frequency of 10 rad/s. Independent hydrogel formulations were mixed for each test.

**B16F10 cell culture.** Luc+ B16F10 cells were purchased from ATCC (ATCC CRL-6475-LUC2). They were cultured in Dulbecco's modified Eagle's media containing 10% FBS, 1% penicillin–streptomycin, and, if Luc+, with 10 μg/mL Blasticidin. Cells were split at a 1:5 ratio every 3 days when approximately 80% confluent. For in vivo experiments, they were injected at passage 3 when 50% confluent. Cells tested negative for mycoplasma using Lonza Mycolert Mycoplasma Detection Kit prior to experiments.

**EG7 cell culture.** EG7 cells were purchased from ATCC. They were cultured in RPMI media containing 10% FBS, 1% penicillin–streptomycin, and 0.05 mM 2-mercaptoethanol. Cell media was supplemented every 2 days and upon confluency (1e6 cells/mL), cells were split by dilution to a seeding density of 100,000 cells/mL. Cells tested negative for mycoplasma using Lonza Mycolert Mycoplasma Detection Kit prior to experiments.

**Short-term viability assay.** Promega CellTiter-Glo 3D Cell Viability Assay was used to characterize the short-term cell viability in different formulation conditions. Cells were seeded between 5000 and 10,000 cells per well in an opaque 96-well plate in 100 μL of media or gel per well. Relative viability was measured after 1 day in culture by adding 100 μL per well of the CellTiter-Glo reagent, mixing for 5 min, allowing the plate to sit for 25 min, and then reading the luminescent signal with a 1 s integration time.

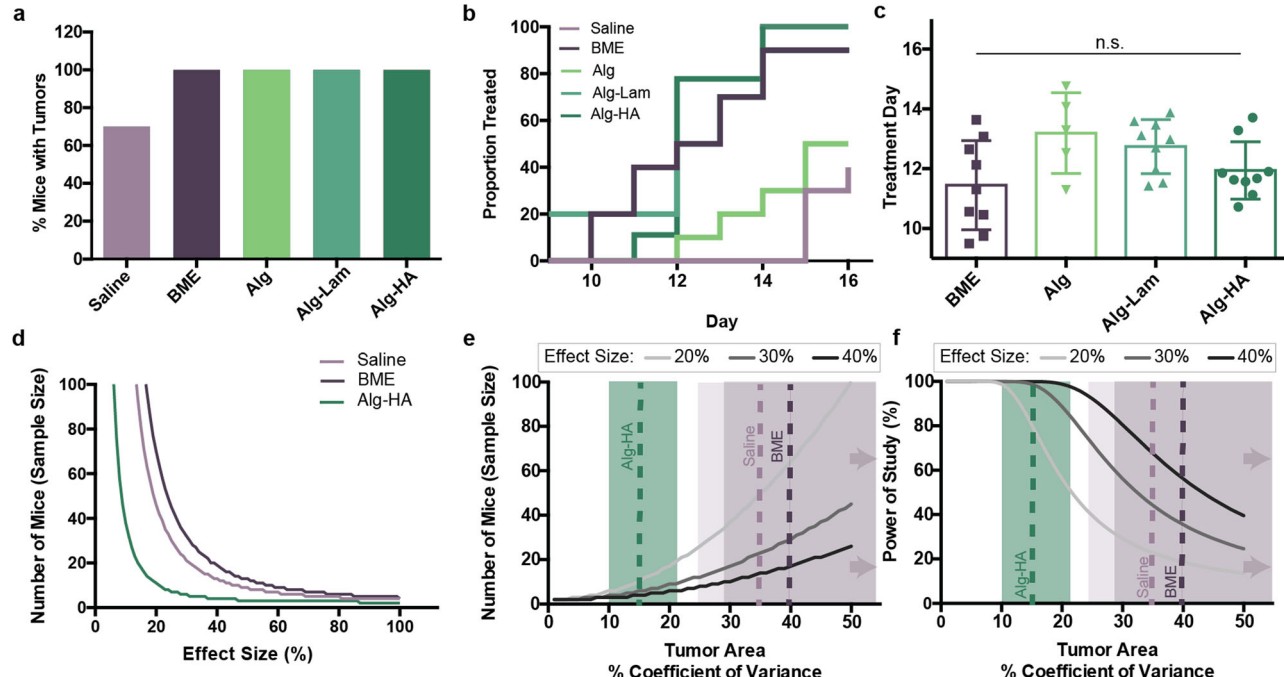

**Fig. 3 Statistical evaluation of tumor growth, treatment, and power. a** Percentage of mice that formed palpable tumors during the 15-day experiment. **b** Treatment day analysis from interpolation assuming exponential growth. Proportion treated from each group over time. **c** Individual points (representing each mouse's treatment day) plotted with average and standard deviation. **d** Power analysis predicting the number of mice per group needed based off the experimental data obtained with 80% power. The Statistical toolbox in Matlab was used for power calculations, specifically the *sampsizepwr* function with two-sided *t*-tests. The average and distribution of the day that each formulation surpassed 100 mm$^2$ were used in the power analysis. **e** Power analyses predicting the effect of the tumor area %CV on the number of mice needed per group to obtain 80% power. Dotted lines represent %CV found from experimental formulations. Shading represents the area encompassing the 95% confidence intervals on the %CV values. The high estimate of the 95% confidence interval for both BME and saline exceed 50%. **f** Power analyses predicting the effect of tumor %CV on the power of the study with 10 mice per group. The dotted colored lines represent the %CV found from experimental formulations. Shading represents the area encompassing the 95% confidence intervals on the %CV values. The high estimate of the 95% confidence interval for both BME and saline exceeds 50%.

**In vivo tumor growth experimental setup**. Luc+ B16F10 cells (Passage 3) were administered to each mouse subcutaneously on the flank in a volume of 50 μL volume containing 400,000 cells using a 1 mL luer-lock syringe with a 21G needle. In all, 400,000 cells were chosen as an intermediate number of cells based on previously reported protocols[9]. Immunocompetent female C57BL/6 mice from Charles River Laboratory (6–8 weeks) were used for all experiments. All experiments followed protocols approved by the Stanford Administrative Panel on Laboratory Animal Care. Each group contained 9 or 10 replicates (*n* = 10 for saline, BME, alginate and Alg–Lam; *n* = 9 for Alg–HA). Experimental groups were blinded and randomized by cage. Hydrogel formulations comprising encapsulated cells were prepared in syringes in the laboratory 30 min prior to inoculation. Cell suspensions were prepared in saline or BME and kept on ice prior to loading into syringes immediately before inoculation. Double the amount of saline volume was prepared to follow established procedures to prevent cell aggregation in the syringe[9].

**Murine tumor growth experiment**. For the first 10 days of in vivo tumor growth experiments, an In Vivo Imaging System (Lago) was used to monitor tumor progression. Firefly luciferin was delivered subcutaneously (150 mg/kg mouse body weight) in 200 μL injection volumes. Luciferase has been found to have immunogenic effects in certain models and was equally used for imaging in all groups for experiments, so side-effects should be uniform among groups[25]. After waiting 10 min, images were recorded with an exposure of 30 s every 5 min for a period of 15 min until maximum flux was reached. Total flux of photons in the tumor region of interest (ROI) was used to quantify tumor growth. Aura imaging software was used to collect and analyze data. Starting on day 10 following tumor inoculation, tumors were measured on each mouse using digital calipers. An area was calculated using length and width caliper measurements. The percent coefficient of variance was calculated as the standard deviation divided by the mean.

**Fit of in vivo imaging data to the Gompertz model**. In vivo imaging data for all groups were fit to the Gompertz model equation (see Supplementary Fig. 4) using GraphPad Prism. Fits were applied on days 3–10 because the administration impacted the initial growth curve on days 1–3. Initial tumor sizes (*X(0)*) for all

groups were extrapolated to time zero for the fit to approximately 100-fold lower than the tumor signal observed on day 3.

**In vivo alginate degradation**. Hydrogels were formulated as previously described though comprising 0.2 wt% Cy7-labeled alginate and 0.8 wt% plain alginate. B16F10 cells were encapsulated and injected in the hydrogels as described previously. Alginate degradation was monitored using an in vivo imaging system and quantified as the total photons in the ROI surrounding the tumor. An exposure of 1 s was used to collect images with an excitation/emission of 720/790 nm.

**Treatment day analysis**. The treatment day was calculated for each mouse by interpolating between two timepoints to when the tumor area reached 100 mm$^2$. Interpolation was performed assuming both exponential and linear tumor growth. The average treatment day and standard deviation for each group were calculated from the individual mice. Mice that required extrapolation before day 9 and after day 16 (tumor reached 100 mm$^2$ before measurement began or after measurements stopped) were removed from the analysis. The following equations describe the linear interpolation approach, where $t_{100}$ represents the day when the tumor reaches 100 mm$^2$, $A_1$ and $A_2$ represent area measurements for corresponding days $t_1$ and $t_2$,

$$t_{100} = \frac{\ln(100) - b}{a}$$
$$\text{where}$$
$$a = \frac{\ln(A_2) - \ln(A_1)}{t_2 - t_1}$$
$$\text{and}$$
$$b = \ln(A_2) - a * t_2.$$

When making the Kaplan–Meier type plot, the treatment day was recorded at the whole number day after a calculated treatment time. The average treatment day was found by averaging individual treatment days. Saline was excluded from this analysis due to the high percentage of mice that never formed tumors.

**Histology and analysis**. Tumors were explanted 15 days after inoculation, immersed in formalin for 72 h, and then 70% ethanol for 48 h. Tumor specimens were sliced and stained with hematoxylin and eosin, CD3, and CD31 by Stanford

Animal Histology Services. All samples were analyzed by a blinded pathologist. For analysis of lymphocyte infiltration, 10 high-power images were taken at all areas of each tumor. Lymphocytes (stained dark brown with a clear circumference) were manually counted in each photo with the FIJI Cell-Counter plug-in.

**Power analysis**. The Statistical toolbox in Matlab was used for power calculations, specifically the *sampsizepwr* function with two-sided *t*-tests. For calculations based off experimental data, the average and distribution of the day that each formulation surpassed 100 mm² were used in the power analysis. Because 30% of the saline tumors did not form at all during the experiment, these mice were excluded from this analysis, leading the saline group to require less mice than the BME group in the power analysis. Here the effect size was calculated as a observed percent change from the formulation average of 100 mm² in 1% intervals to predict the number of mice needed. For calculations predicting the effect of the coefficient of variance on the number of mice, the variance was calculated as a percent change from 100 mm² in 1% intervals with a constant 80% power. For calculations predicting the effect of variance on the power of the study, the variance was calculated as a percent change from 100 mm² in 1% intervals with a constant 10 mouse per group sample size. Example code is shown in the Supplemental Information. Code can be shared upon reasonable request.

**Statistical treatment to find confidence intervals on %CV**. Confidence intervals in %CV values were calculated using Vangel's modification to McKay's method, which assumes the data are approximately normal and starts giving less valuable estimates as the %CV exceed 33%[26]. To summarize this calculation, the lower (lcl) and upper limits (ucl) were calculated as

$$\text{lcl} = \frac{K}{\sqrt{\left(\frac{u_1+2}{n}-1\right)*K^2+\frac{u_1}{n-1}}},$$

$$\text{ucl} = \frac{K}{\sqrt{\left(\frac{u_2+2}{n}-1\right)*K^2+\frac{u_2}{n-1}}},$$

where

$$u_1 = \chi^2_{1-\alpha/2,n-1}$$

and

$$u_2 = \chi^2_{\alpha/2,n-1}.$$

$K$ represents a sample %CV, $n$ is the number of samples, and $\chi^2_\alpha$ represents the chi-squared distribution at the designated confidence $\alpha$.

**Statistics and reproducibility**. All error bars represent the standard deviation unless otherwise specified.

**Reporting summary**. Further information on research design is available in the Nature Research Reporting Summary linked to this article.

## Data availability

All data supporting the results in this study are available within the Article and its Supplementary Information. Manuscript raw data can be found at https://figshare.com/articles/dataset/DataCompiled_xlsx/14813880/1[27]), and all other data are available from the corresponding author upon reasonable request.

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

## Author contributions

A.K.G. and E.A.A. conceived of the idea and planned experiments. A.K.G. performed experiments. S.C. helped with experimental planning and setting up experiments. R.A.B. and C.L.M. aided in analysis of experimental data and statistical methods. E.C.G. helped perform experiments. All authors contributed to the final manuscript.

## Competing interests

E.A.A., A.K.G., and S.C. are listed as inventors on a provisional patent application (63/094716) filed by Stanford University describing the technology reported in this manuscript. The remaining authors declare no competing interests.
