## [Transparent Peer Review File · Communications Biology]

Reviewers' comments:

Reviewer #1 (Remarks to the Author):

Summary

The authors report the development of a tumor inoculation method using self-assembled hydrogels to reproducibly generate tumors in subcutaneous murine tumor model. The authors identified a very fundamental issue in pre-clinical models of cancer research. The authors present self-assembling alginate gels functionalized with HA and laminin to support the tumor cells. A robust comparative analysis performed by the authors show that the tumour formation was much more reproducible in gel formulation treatment groups as compared to saline and BME. The authors also showed impact of the improved model on the reduction of cohort size and increase in the power of the study design.

Major comments

1. The authors should comment on the impact of the process of injection and pre-gelation in the syringe on the inoculation and reproducibility for other cancerous models that might require an in situ gelation.
2. What was the rationale in determining the HA and laminin concentrations for the study? Was a dose-optimization study performed?
3. Figure 2 (a:e): The graphs of Cv should be presented as a comparison between different treatments over a time line to give a clear comparison as opposed to splitting the information into five graphs.
4. The authors should comment on the observation of H&E images where saline and alginate conditions seem very comparable or use different representative images.
5. In supplementary fig. 10, the authors should indicate if there was a significant difference between the observed standard deviations of all treatment groups.
6. Mode and volume of delivery should be indicated clearly in the manuscript.
7. Figure 3: (c) The stats on this graph are missing and should be included to give a clear significance of the results. (d, e, f) The authors should include the Lam-Alginate and Alginate only groups too or explain the absence of those groups.
8. The authors should include a comment on the impact of the alginate system used for tumor inoculation in terms of screening drugs/ therapeutic interventions for cancer research. It might have a 'cocooning' impact to the applied therapeutics.
9. Was the in vivo degradation of the alginate system investigated? It might be of relevance to give an insight on the treatment regimens to be tested using these models.
10. Statistical analysis and significance should be included in all the relevant supplementary figures (eg. Fig 4, Fig 10)
11. The authors should provide an explanation of the H&E images in the figure legends for supplementary figures 5-9 and explain the differences that they want to be pointed out. For the amorphous material, it is recommended to highlight the same using asterisk or arrows.

Minor comments

1. All figure legends should be modified to be a stand-alone and reflect the potential results from analysis.
2. Figure 3: Indicate the statistical analysis in figure caption.
3. All the figure legends in Supplementary figures should be stand alone and should reflect the results that the authors are using them to support.

Reviewer #2 (Remarks to the Author):

In this study, the authors developed a hydrogel for transplantation of cancer cells in mouse models. For demonstration, the B16-F10 tumor cells admixed with gel were implanted subcutaneously into C57BL/6 mice, resulting in more uniform tumor growth.

In preclinical modeling, "low reproducibility" should not be confused with "heterogeneity". For example, tumor responses in a mouse model can be heterogeneous among individual mice, but the outcome can still be reproducible to validate the therapeutic efficacy (e.g. by Kaplan-Meier analysis). In contrast, tumor growth can be all uniform in each study, but the therapeutic efficacy could be inconsistent between studies. Generally speaking, reproducibility between studies is determined by the accumulation of variation of driving factors. The major players are injected cells (number and condition in culture), hosts (age, gender and health), the housing condition of hosts (stress and microbiome), and quality of therapeutic agents.

The cell matrix materials, such as collagen gel, fibrogel, and Matrigel can not only protect cells from anoikis and microenvironment stress, but also provide growth factors after degradation by protease. In general, they can enhance tumor-take rate in hosts. However, these materials may also select or out-select specific cellular characteristics, skewing the study outcome or even making the model not matching the real disease anymore. After all, the most important issue in preclinical studies is to justify what is the "real" system to be modeled exactly (cancer subtype, immune response, specific signaling pathway, etc.)

With all these notions, several factors need to be characterized before the hydrogel can be used in preclinical studies:

1. B16-F10 can be subcutaneously injected at the number of 1 million in C57BL/6 mice to reach acceptable tumor-take rate. 400,000 cells were injected in this study. It should be noted if this is for the purpose to test the capacity of the hydrogel.

2. What is "HA" in Alg-HA?

3. Luciferase is immunogenic and can cause immune response in immunocompetent mice (<https://journals.plos.org/plosone/article?id=10.1371/journal.pone.0109956>). Alg gels significantly enhanced cell survival upon injection (Supplementary Figure 3), but resulted in smaller tumor sizes later. How Alg affects angiogenesis and local immune response need to be investigated (see point 4 and 5 below). Also, Growth kinetics of Supplementary Figure 3 needs to be analyzed (e.g. by fitting it to Gompertzian model) to identify the step of tumor growth influenced by hydrogel.

4. Is Alginate itself immunogenic? Could that be the cause that tumor growth was better in Alg-Lam and Alg HA than Alg only?

5. Assuming the histology images in the Supplementary Fig. 5-9 are well representative, tumors growing in different matrix materials had significant difference in angiogenesis, necrosis, and pigmentation. At day 10, Alg gels gave better tumor-take rate. However, at Day 15, tumors in Alg gels are more uniform but significantly smaller than those in Matrigel or even saline control. Could Alg gels inhibit angiogenesis? Staining and quantitation of regular immune (CD3, CD4/CD8 and macrophage markers) and angiogenesis (CD31, CD34) markers are needed to justify such effects.

6. Why was tumor size measured by area in Fig. 2 but by volume in Supplementary Fig. 4?

7. Alg gels may have great value in improving patient-derived xenograft (PDX) models, whose tumor-take rates are notoriously variable. If a PDX is not readily available for testing, I suggest the authors to test them in the transplantation of NCI-H358 and NCI-H2030 NSCLC cell lines in nude mice or NSG mice. Both are the well-known difficult grower in transplantation models. The tests will greatly promote the value of Alg gels in preclinical models.

Reviewer #1 (Remarks to the Author):

The authors report the development of a tumor inoculation method using self-assembled hydrogels to reproducibly generate tumors in subcutaneous murine tumor model. The authors identified a very fundamental issue in pre-clinical models of cancer research. The authors present self-assembling alginate gels functionalized with HA and laminin to support the tumor cells. A robust comparative analysis performed by the authors show that the tumour formation was much more reproducible in gel formulation treatment groups as compared to saline and BME. The authors also showed impact of the improved model on the reduction of cohort size and increase in the power of the study design.

We thank the reviewer for their supportive comments. We have responded to the comments outlined below in our revised paper. In particular, we have added additional statistical analysis and immune staining, and performed an *in vivo* experiment measuring hydrogel degradation.

Major comments

1. The authors should comment on the impact of the process of injection and pre-gelation in the syringe on the inoculation and reproducibility for other cancerous models that might require an in situ gelation.

This technique is applicable to most solid-tumor models. The pre-gelation of this hydrogel does not affect the injectability of the gel and allows for rapid self-healing once injected. It should remain applicable to all applications, even those currently relying on in situ gelation, and should only improve upon current inoculation strategies.

2. What was the rationale in determining the HA and laminin concentrations for the study? Was a dose-optimization study performed?

The concentrations were chosen to be the maximum possible amount we could add to formulate the hydrogel with its other components. We have clarified these concentrations and considerations in the manuscript. For example, we dosed 50 ug Laminin per mouse as it is only produced commercially or in-house in labs at concentrations of 1-2 mg/mL, so we opted to use the highest concentration we could formulate into the gels. In contrast, we dosed 1 mg HA per mouse. This polymer (MW1.5 Mkda) could not be added at higher quantities because it is highly viscous at high concentrations and alters the formulation injectability at higher concentrations.

We have clarified the rationale for the dosing decisions in the manuscript and clearly explained that some concentrations of additives must be limited to retain the properties of the hydrogel and the injectability of the formulations. Further studies could be performed with lower concentrations of the additives, but these would likely show results similar to the plain alginate (Alg) control group.

3. Figure 2 (a:e): The graphs of CV should be presented as a comparison between different treatments over a time line to give a clear comparison as opposed to splitting the information into five graphs.

We have added a figure (Supplementary Fig. 7, reproduced below) with all the %CV values over time in the supplementary information to display the data more clearly and show more direct comparisons between groups. We have also added grey lines to guide the eye in Figure 2 to help the reader make comparisons.

Supplementary Figure 7: %CV of tumor area for all groups: saline, BME, Alg, Alg-Lam and Alg-HA. Same data as Fig 2a plotted on one graph to show direct comparisons. Saline exceeded an 100 %CV on Days 10 and 13.

4. The authors should comment on the observation of H&E images where saline and alginate conditions seem very comparable or use different representative images.

The H&E histology is meant to show that the conditions actually are highly comparable through the series of consistent images, and that we do not observe any changes in the resulting tumor morphology in our hydrogel groups from those generated using standard protocols. We have further clarified this finding in the manuscript.

5. In supplementary fig. 10, the authors should indicate if there was a significant difference between the observed standard deviations of all treatment groups.

The calculated treatment day was not statistically different between groups, which we have now highlighted specifically in Fig 3c in the revised manuscript. In Supp. Fig. 16 (shown below; previously Supp. Fig. 10), the standard deviations between groups vary between day 1 and day 2. Indeed, we observe a 100% change (which is a rather large difference) that becomes even more important when put in context of preclinical experimentation. When the hydrogel is used, the treatment day for all the mice spans over only 48 hours, meaning researchers can start all treatments and perform examinations at similar timepoints. In contrast, when BME is used, the treatments for all the mice span over 4 days, meaning researchers must spread treatments and analysis over a much broader timeframe, posing a significant logistical challenge (4 days is a significant timespan when tumors form starting only 10 days after inoculation).

Supplementary Figure 16: Standard deviation of treatment day for each formulation. Standard deviations from the averages shown in Fig 3c.

6. Mode and volume of delivery should be indicated clearly in the manuscript.

We have further clarified in the manuscript that we delivered 50 μ L of saline, BME or hydrogel containing 400,000 cells subcutaneously in the flank using a 1 mL luer-lock syringe with 21 G needle.

7. Figure 3: (c) The stats on this graph are missing and should be included to give a clear significance of the results. (d, e, f) The authors should include the Lam-Alginate and Alginate only groups too or explain the absence of those groups.

To more thoroughly evaluate the significance between groups, we have produced a new supplementary figure and modified Figure 3 in the manuscript. As the coefficient of variance is itself a type of statistical analysis, statistical analysis of various coefficients of variance is a topic of current research. We have used a relatively new method to perform statistical analysis on our %CV values to include 95% confidence intervals on the %CV values reported in Fig 3e and 3f (reproduced below). Confidence intervals were calculated using Vangel's modification to McKay's theory on the coefficient of variance. We have included the citation and mathematical calculations in the methods section. We have also included a new Supplementary Figure 17 (reproduced below) with the statistical analysis.

In this analysis, we included the best formulation from our empirical studies (Alg-HA), along with saline (the most commonly used method) and BME (the gold standard). We believe that since this Alg-HA treatment group is the most relevant, it was fitting to reduce the number of groups examined in the subsequent analysis to reduce redundancy, complexity and to more clearly demonstrate the implications of these novel methods. We have clarified this rationale in the text.

Supplementary Figure 17: a-b, Power analysis predicting the number of mice per group needed based off the experimental data obtained with 80% power. The dashed lines and shading represent the high and low number estimate of mice given the high and low 95% confidence interval values on the input coefficient of variance for these power calculations. Confidence intervals on coefficient of variation were calculated using Vangel's modification on McKay's theory on coefficient of variance. The Statistical toolbox in Matlab was used for power calculations, specifically the *sampsizepwr* function with two-sided t tests. The average and distribution of the day that each formulation surpassed 100 mm² were used in the power analysis.

Figure 3 e, Power analyses predicting the effect of the tumor area %CV on the number of mice needed per group to obtain 80% power. Dotted lines represent %CV found from experimental formulations. Shading represents the area encompassing the 95% confidence intervals on the %CV values (Vangel's method). The high estimate of the 95% confidence interval for both BME and saline exceed 50%. **f**, Power analyses predicting the effect of tumor %CV on the power of the study with 10 mice per group. The dotted colored lines represent the %CV found from experimental formulations. Shading represents the area encompassing the 95% confidence intervals on the %CV values. The high estimate of the 95% confidence interval for both BME and saline exceed 50%.

8. The authors should include a comment on the impact of the alginate system used for tumor inoculation in terms of screening drugs/ therapeutic interventions for cancer research. It might have a 'cocooning' impact to the applied therapeutics.

We inject a very small amount of alginate and have found through *in vivo* degradation studies that this alginate remains intact for only the first few days after inoculation (see Supplementary Figure 6 reproduced below). Moreover, histology studies indicate that histomorphology is comparable among all inoculation strategies evaluated. These observations lead us to conclude that there is not a "cocooning" effect that may affect drug or therapeutic testing outcomes because the tumor matrix is remodeled by the cells. We have included additional commentary on these findings in the revised manuscript.

9. Was the *in vivo* degradation of the alginate system investigated? It might be of relevance to give an insight on the treatment regimens to be tested using these models.

We have performed an additional experiment to investigate the degradation of the alginate gels encapsulating the B16F10 cells. We chemically labelled alginate with a fluorescent Cy7 dye and monitored the presence of the alginate in the subcutaneous space over time using IVIS imaging. We found that the alginate in all hydrogel groups degrades over the course 10-12 days with a half-life of around 4 days for all groups. This timescale of degradation aligns with the growth of the B16F10 cells into robust tumors, suggesting there should not be significant effects of the alginate being present during treatment, which would only occur after complete degradation of the alginate and establishment of the tumors. We have added a new Supplementary Figures 5 and 6 (reproduced below), and have updated the methods and related discussion accordingly.

Supplementary Figure 6: **a**, *In vivo* degradation of fluorescent alginate measured using an *in vivo* imaging system for all hydrogel groups. B16F10 cells were encapsulated and co-injected in all groups as in previous studies. **b**, Average total flux and standard deviation from the region of interest surrounding the injection over time. A one-phase exponential decay was fitted to each curve and a half-life and standard deviation was computed for each group as shown on the graph.

10. Statistical analysis and significance should be included in all the relevant supplementary figures (eg. Fig 4, Fig 10)

We have now included further statistical analysis for all figures and supplementary figures. Supplementary Figure 4 has been removed because it was redundant. Statistics have been added to Figure 3c (reproduced below, corresponding to Supplementary Fig 10).

Additionally, the most relevant p values were added to Fig 2 comparing the hydrogel groups to the controls (reproduced below). We could not include all 30 p values on the figure itself due to space constraints, so we included Supplementary Table 1 with all associated p values for tumor areas from the data shown in Figure 2 of the manuscript. The table is reproduced below.

Figure 3 c, Individual points (representing each mouse's treatment day) plotted with average and standard deviation.

Figure 2. Syngeneic tumor model evaluation. a-e, Tumor area measurements over time following inoculation with various formulations containing encapsulated B16F10 cells. Average tumor area and standard deviation, and the corresponding resulting %CV. Supplementary Table 1 displays all p values. Saline exceeded an 100 %CV on Days 10 and 13. Grey lines included to guide the eye for better comparison.

Supplementary Table 1: P-values calculated with two-sided t-tests from analysis of tumor areas for all groups (Fig 2).

Group Pair	Day 10	Day 13	Day 15
Saline, BME	0.0002	0.0011	0.0045
Saline, Alg	<0.0001	0.0239	0.2778
Saline, Alg-Lam	<0.0001	0.0014	0.0290
Saline, Alg-HA	<0.0001	0.0015	0.0027
BME, Alg	0.5848	0.0163	0.0076
BME, Alg-Lam	0.9994	0.2423	0.0978
BME, Alg-HA	0.2800	0.2251	0.3943
Alg, Alg-Lam	1.0000	0.0282	0.0501
Alg, Alg-HA	1.0000	0.0113	0.0003
Alg-Lam, Alg-HA	0.1637	0.9485	0.1133

11. The authors should provide an explanation of the H&E images in the figure legends for supplementary figures 5-9 and explain the differences that they want to be pointed out. For the amorphous material, it is recommended to highlight the same using asterisk or arrows.

Additional explanation has been included in the figure captions, and arrows have been added to point out the mentioned amorphous materials. The primary inclusion of these H&E images was to show further replication for Figure 3 from the main manuscript. All of the images show that tumors were histomorphologically comparable across all groups evaluated, so there aren't differences to highlight.

Minor comments

1. All figure legends should be modified to be a stand-alone and reflect the potential results from analysis.

All figure captions were reviewed and additional information was added to give all figures the ability to stand-alone.

2. Figure 3: Indicate the statistical analysis in figure caption.

We have further explained the statistical analysis and methods in the Figure 3 caption.

3. All the figure legends in Supplementary figures should be stand alone and should reflect the results that the authors are using them to support.

All supplementary figure captions were reviewed and additional information was added to ensure that all figures are able to stand alone.

Reviewer #2 (Remarks to the Author):

In this study, the authors developed a hydrogel for transplantation of cancer cells in mouse models. For demonstration, the B16-F10 tumor cells admixed with gel were implanted subcutaneously into C57BL/6 mice, resulting in more uniform tumor growth. In preclinical modeling, “low reproducibility” should not be confused with “heterogeneity”. For example, tumor responses in a mouse model can be heterogeneous among individual mice, but the outcome can still be reproducible to validate the therapeutic efficacy (e.g. by Kaplan-Meier analysis). In contrast, tumor growth can be all uniform in each study, but the therapeutic efficacy could be inconsistent between studies. Generally speaking, reproducibility between studies is determined by the accumulation of variation of driving factors. The major players are injected cells (number and condition in culture), hosts (age, gender and health), the housing condition of hosts (stress and microbiome), and quality of therapeutic agents.

The cell matrix materials, such as collagen gel, fibrogin, and Matrigel can not only protect cells from anoikis and microenvironment stress, but also provide growth factors after degradation by protease. In general, they can enhance tumor-take rate in hosts. However, these materials may also select or out-select specific cellular characteristics, skewing the study outcome or even making the model not matching the real disease anymore. After all, the most important issue in preclinical studies is to justify what is the “real” system to be modeled exactly (cancer subtype, immune response, specific signaling pathway, etc.)

With all these notions, several factors need to be characterized before the hydrogel can be used in preclinical studies:

We thank the reviewer for their insightful comments. We have responded to all comments below and revised our manuscript accordingly. In particular, we have added new statistical analysis, additional immune staining, and performed an *in vivo* experiment measuring hydrogel degradation over time.

1. B16-F10 can be subcutaneously injected at the number of 1 million in C57BL/6 mice to reach acceptable tumor-take rate. 400,000 cells were injected in this study. It should be noted if this is for the purpose to test the capacity of the hydrogel.

We chose 400,000 cells because the literature suggests between 100,000 and 1,000,000 cells was a common for inoculation protocols. We opted for an intermediate value according to Overwijk et al. (Current Protocols 2001). We did not intentionally aim to test the capacity of the hydrogel, but our results may show that our hydrogel is more robust to lower cell numbers compared to PBS controls. We have explained our rationale in greater detail in the methods.

2. What is “HA” in Alg-HA?

HA refers to Hyaluronic Acid. We have further clarified this naming convention in the manuscript and included additional rationale for the dose of HA used in these materials.

3. Luciferase is immunogenic and can cause immune response in immunocompetent mice (<https://journals.plos.org/plosone/article?id=10.1371/journal.pone.0109956>). Alg gels significantly enhanced cell survival upon injection (Supplementary Figure 3), but resulted in smaller tumor sizes later. How Alg affects angiogenesis and local immune

response need to be investigated (see point 4 and 5 below). Also, Growth kinetics of Supplementary Figure 3 needs to be analyzed (e.g. by fitting it to Gompertzian model) to identify the step of tumor growth influenced by hydrogel.

The use of luciferase-positive cells is highly common and widely accepted for measuring tumor growth in live animal models. We dosed the same amount of luciferase-positive cells in all groups (including hydrogels and controls), so any effects that may arise due to the expression of luciferase should be equal among groups. We have included an additional note in our revised manuscript that luciferase can cause immune responses and included the citation noted by the reviewer.

All hydrogel groups showed enhanced cell survival upon injection at early time points, but only the plain alginate group exhibited slowed tumor growth at later timepoints when compared to the Alg-Lam, Alg-HA and BME groups. These findings suggest that it was likely the difference in formulation (e.g., lack of inclusion of a biomolecule such as Laminin or Hyaluronic Acid) that caused slowed tumor growth rather than the presence of more luciferase.

Growth kinetics from Supplementary Fig 3 were analyzed using the Gompertz model as suggested by the reviewer and are now included in Supplementary Fig 4 (reproduced below). This analysis helps demonstrate that the hydrogels do not hinder proliferation of the cells

Supplementary Figure 4: *In vivo* imaging fit to Gompertz model. Parameter α that represents the proliferation constant, parameter K that represents the carrying capacity, $X(0)$ represents the initial size, t represents time (days). **a**, Tumor luminescence fits to Gompertz equation after initial delivery phase (starting on Day 3, see methods for details). **b**, Parameter α in the Gompertz model and standard error calculated from the fit for all groups. **c**, Parameter K from the Gompertz model and standard error calculated from the fit for all groups.

compared to saline, and may indeed improve the carrying capacity (e.g., longterm growth size of the tumor based on nutrient balance). We have added this figure and associated discussion and methods to the revised version of the manuscript.

4. Is Alginate itself immunogenic? Could that be the cause that tumor growth was better in Alg-Lam and Alg HA than Alg only?

Correctly sterilized alginate has been shown to have minimal immunogenicity and good biocompatibility, especially if it has higher G content compared to M (<https://www.ncbi.nlm.nih.gov/pmc/articles/PMC3223967/>). We purchased our Alginate from a highly pure source with high G content (Pronova LVG UP, NovaMatrix) and performed appropriate sterilization procedures prior to administration in mice. Our experimental findings, specifically the histological analysis, suggest that the alginate is non-immunogenic. Since the Alg-Lam and Alg-HA formulations contained the same amount of alginate as the Alg (only alginate) formulation, the Alg-Lam and Alg-HA formulations exhibited similar growth rates to controls, and since all groups exhibited similar histomorphology, we believe it is unlikely the alginate itself was immunogenic in our experiments. If the alginate were indeed immunogenic, we would have expected to observe reduced tumor growth in all hydrogel groups. We believe tumor growth was improved in the Alg-Lam and Alg-HA groups because laminin and hyaluronic acid are highly tumorigenic molecules. We have added a citation to the reference above and added a statement regarding alginate immunogenicity to the text of the revised manuscript.

5. Assuming the histology images in the Supplementary Fig. 5-9 are well representative, tumors growing in different matrix materials had significant difference in angiogenesis, necrosis, and pigmentation. At day 10, Alg gels gave better tumor-take rate. However, at Day 15, tumors in Alg gels are more uniform but significantly smaller than those in Matrigel or even saline control. Could Alg gels inhibit angiogenesis? Staining and quantitation of regular immune (CD3, CD4/CD8 and macrophage markers) and angiogenesis (CD31, CD34) markers are needed to justify such effects.

We have performed further staining analysis to investigate these differences. CD3 staining was performed and lymphocyte infiltration was investigated (n=3-4 tumors per group). The number of lymphocytes was highly variable tumor-to-tumor and did not show clear trends between groups. Of all the groups, the alginate-only (Alg) group showed the least amount of lymphocyte infiltration. Supplementary Fig 15 (shown below) was included and described in the text.

Supplementary Figure 15: a, Number of lymphocytes per tumor found in 10 high power fields (40x magnification) for all experimental groups. b, Representative high power images of CD3 stained tumors where lymphocytes are stained dark brown. Scale bar represents 50 μ m.

A blinded pathologist also examined CD31 staining of tumors in all groups (n=3-4) and a new Supplementary Fig 14 (shown below) has been included in the revised manuscript. No differences were noted between groups; however, it was difficult to perform a quantitative analysis due to the background melanin present in B16F10 tumors. Instead, to assess angiogenesis, a blinded pathologist performed a quantitative analysis of the H&E staining by quantifying % necrosis for each tumor. This analysis is included in Supplementary Figure 13 (shown below) and described in the text of the revised manuscript.

Supplementary Figure 13: % Necrosis as determined by a blinded pathologist for each tumor in all experimental groups.

The % necrosis was highly heterogenous tumor-to-tumor and did not show consistent differences between groups, except that the alginate-only (Alg) group exhibited consistently lower % necrosis. However, this observation may be correlated to the sizes of the tumors because the Alg group also showed consistently smaller tumors. Indeed, the Alg group tumors were likely the smallest rather because they did not have any additive tumorigenic components (like the Laminin or HA) to trigger B16F10 cell proliferation.

The fact that the hydrogel groups showed comparable necrosis and lymphocyte infiltration to PBS and BME controls suggests that our approach does not change the tumor environment in any meaningful way and that the tumors formed in hydrogels are consistent with these commonly used approaches.

Supplementary Figure 14: Representative CD31 staining from each group. No significant differences were observed by a blinded pathologist. Background melanin made clear analysis challenging. Scale bar represents 250 μ m.

6. Why was tumor size measured by area in Fig. 2 but by volume in Supplementary Fig. 4?

The volume data shown in Supplementary Fig 4 was calculated from the tumor areas shown in Figure 2 and is therefore redundant. We have decided to remove Supplementary Fig 4 to limit redundancy and improve clarity.

7. Alg gels may have great value in improving patient-derived xenograft (PDX) models, whose tumor-take rates are notoriously variable. If a PDX is not readily available for testing, I suggest the authors to test them in the transplantation of NCI-H358 and NCI-H2030 NSCLC cell lines in nude mice or NSG mice. Both are the well-known difficult grower in transplantation models. The tests will greatly promote the value of Alg gels in preclinical models.

Due to COVID-19 lab restrictions, we are incapable of completing this experiment in the near future, but plan to complete it when lab access expands. We thank you for your consideration.

REVIEWERS' COMMENTS:

Reviewer #2 (Remarks to the Author):

1. In the rebuttal letter (page 10 and 11), Supplementary Fig. 13 and 15 were inserted incorrectly.
2. Please explain how to read Supplementary Fig. 1 in the legend.
3. The heterogeneity of tumor growth in vivo can be caused by (1) the capacity of cancer cells to grow and adapt to the environment; (2) the variation in the hosts, including angiogenesis, immune response. The authors tended to attribute the improvement by using Alg-based hydrogels to (1), including the growth-stimulating biomolecules provided by Alg-Lam and Alg-HA. However, the analysis is inconclusive. Such difference should be further characterized, because it may impact the outcome of preclinical study very differently. To answer this question, two studies are required: (a) an in vitro test for cancer cells growth in different matrix or hydrogels; (b) a preclinical study to compare response of cancer cells implanted with different matrix/hydrogels to a drug.

Understandably, a big scale of preclinical study is very difficult to organize under the current pandemic situation. Therefore, the in vitro study should be sufficient to address (1), and the results should be discussed in the manuscript. This would be a very important information, because it will determine the type of preclinical study the hydrogel can be applied (e.g. chemotherapy vs. immunotherapy).